# Maltese as a merger of two worlds: A cross-language approach to phonotactic classification

Jessica Nieder[1]*, Fabian Tomaschek[2]

1 Department of Linguistics, Heinrich-Heine-Universität Düsseldorf, Düsseldorf, Germany, 2 Department of General Linguistics, Eberhard-Karls-Universität Tübingen, Tübingen, Germany

☉ These authors contributed equally to this work.

* nieder@phil.hhu.de

**Data Availability Statement:** The data underlying the results presented in the study are available from https://osf.io/r9k5n/.

**Funding:** This research was funded by the Deutsche Forschungsgemeinschaft (Research Unit FOR2373 'Spoken Morphology'). Project VI 223/3-

## Abstract

Maltese is a prime example of a language that emerged through extensive language contact, joining the two linguistic worlds of Semitic and Italo-Romance languages. Previous studies have shown this shared origin on the basis of hands-on comparative methods. However, such approaches may be biased by the researchers perspective and the selected material. To avoid this bias, we employed a naive computational method that classifies words on the basis of their phonotactics. Specifically, we trained a simple two-layer neural network on Tunisian and Italian nouns, i.e. the languages that Maltese emerged from. We used the trained network to classify Maltese nouns based on their phonotactic characteristics as either of Tunisian or of Italian origin. Overall, the network is capable of correctly classifying Maltese nouns as belonging to either of the original languages. Moreover, we find that the classification depends on whether a noun has a sound or broken plural. By manipulating the segment identity in the training input, we found that consonants are more important for the classification of Maltese nouns than vowels. While our results replicate previous comparative studies, they also demonstrate that a more fine grained classification of a language's origin can be based on individual words and morphological classes.

## Introduction

### Overview

By means of comparative methods, research has established that languages such as French and Italian belong to the Romance language family, whereas Hebrew and Arabic belong to the Semitic language family. The classification is based on two sources of evidence. The first direct source of evidence is that for some languages a phylogenetic predecessor is documented. The second—indirect—source of evidence is a comparison between languages on the basis of syntactic, morphological and phonological characteristics: languages that belong to the same language family share many linguistic characteristics while they differ in these characteristics from those languages that belong to another family [1]. Once in a while, however, through

2: Prof. Ruben van de Vijver (PI), Dr. Jessica Nieder (PhD/Postdoc) Project BA 3080/3-2: Prof. Harald Baayen (PI), Dr. Fabian Tomaschek (Postdoc) The funders had no role in study design, data collection and analysis, decision to publish, or preparation of the manuscript.

**Competing interests:** The authors have declared that no competing interests exist.

intensive language contact, languages emerge that share a high number of features with two (or more) language families. One of such languages is Maltese. Numerous studies have demonstrated that the grammatical foundation of Maltese is Arabic, i.e. a Semitic language. By contrast, its lexicon, i.e. its verbs, nouns and adjectives, exhibits morphological and phonotactic characteristics of both Semitic and Italo-Romance languages [2–4].

Studies on Maltese have mostly employed a traditional approach. Words and their linguistic characteristics were compared manually using a 'pen and paper' method [4]. Clearly, such a manual method restricts the amount of words and detail that can be investigated and also may be biased by the selected material. This is why the manual method has recently been supplemented by the use of computational methods that do not only reduce the processing time but also increase the size of data that can be investigated. Therefore, the first goal of the present study is to establish that Maltese, and its relation to other language families, can indeed be investigated by means of computational methods.

In doing so, we employ a less biased method for the comparison of Maltese with Semitic and Italo-Romance languages and simultaneously increase the amount of data investigated. Accordingly, our study introduces a computational method based on phonotactic classification to the study of the origin of Maltese.

Furthermore, previous studies employing computational methods were used to reconstruct ancestral forms and phonological characteristics of dead languages [5, 6], or to identify a language's membership to a language family [7–12]. Such an approach is able to tell us a lot about the genetic relations between languages. However, only a few studies have employed this approach to study languages in contact [13]. Moreover, since studies using the computational approach typically focus on classifying a language as belonging to either one or the other family, they do not pay much attention to the fine details of classification within a language itself. Specifically, the question arises how the classification varies from word to word.

To answer this question, we trained a simple neural network to classify nouns from Tunisian Arabic or Italian as either Semitic or Non-Semitic on the basis of their phonotactics. Subsequently, we used the trained network to classify individual Maltese singular and plural nouns as belonging to either of these two language families. We expected that our classification would vary in relation to two sources of variability. The first source is the origin of the nouns as established by traditional approaches. The second source of potential variability is the nouns' morphological class which depends on the way the plural is formed. Most of Maltese nouns with a Semitic origin have a non-concatenative plural, while most of the nouns with a Non-Semitic origin have a concatenative plural. Nevertheless, both morphological classes contain nouns that show a pluralization pattern from the other class. In other words: some Semitic nouns have a concatenative plural and some Non-Semitic nouns have a non-concatenative plural.

This brings us to the second goal of the present study: We wanted to investigate how the classification of nouns as either Semitic or Non-Semitic varies on the basis of their origin and on the basis of their morphological class. To anticipate the results, we find that Maltese nouns can be classified with a relatively high accuracy as belonging to either of the two language families. Moreover, the probability of being classified as either Semitic or Non-Semitic depends on the (established) origin of the noun as well as on whether it has a concatenative or non-concatenative morphology. In the following section, we present additional background on Maltese. Subsequently, we present our hypotheses for the present study.

## Maltese history

The history of Maltese is intimately connected to the geographical location of the islands of Malta. The islands are located in the Mediterranean sea between two regions representing two

different cultural and linguistic worlds. Sicily is located roughly 100 km to the north, Tunisia is located roughly 300km to the south [14]. This geographical location between Sicily and Tunisia resulted in a long history of economical and political relations and, unsurprisingly, to language contact between the Italo-Romance languages (Sicilian Italian) and the Semitic languages (Maghrebi Arabic vernaculars). The first contact with Maghrebi Arabic vernaculars started after the Arab conquest of Malta in 870 CE and its influence on Maltese as well as its role in the foundation of the Maltese grammatical system is well documented in the literature [14–17]. The influence of Sicilian Italian on Maltese already started while Malta was settled by Arabs due to their ongoing contact with Sicily [17]. Its strongest influence on Maltese, though, is attributed to the time period of the Middle Ages, when the islands of Malta became part of the kingdom of Sicily [18]. During this time, the influence is reflected by the many borrowed words, as well as morphological structures, while the Arabic grammatical foundation remained the same. Nowadays, Maltese is the only Semitic language spoken by roughly 520.000 native speakers (according to the preliminary census report of 2021 available at https://census2021. gov.mt/results/) in the European Union that is written in a Latin alphabet and serves as a national and official language.

The impact of the two linguistic worlds on the Maltese grammatical system is mostly reflected in the interplay between Maltese phonotactics and its morphology, which is why [15] beautifully states that Maltese constitutes

> "[. . .] a single system, in which Romance and English vocabulary items are manipulated within a framework of Semitic origin, while the Semitic framework itself is influenced by the borrowings."

The complex interrelation between Semitic and Non-Semitic structures becomes mostly apparent in Maltese nouns—the targets of the present study. Maltese nouns can be grouped into two categories. The first group of nouns is constituted by those that show a non-concatenative plural inflection. Most of these nouns were incorporated into the language from Maghrebi Arabic vernaculars [16, 17], which is why they can be regarded as inherently Semitic.

The second group of nouns shows a concatenative plural inflection. This group of nouns largely originates from extensive borrowings from Italo-Romance languages, mostly Sicilian. In addition—though not of importance for the present study—some borrowings originate from English due to its colonialization by the British Empire.

With the history of Maltese in mind, we discuss details of the noun system in the next section.

## Maltese plural system

As we have stated above, language contact in the history of Maltese resulted in a relative complex morphological system to pluralize nouns [19–21]. On the one hand, there are 12 concatenative suffixes which form the so-called 'sound plurals' e.g. *pjaneta—pjaneti* 'planets'. On the other hand, there are 11 different non-concatenative phonotactic changes, which form the so called 'broken plurals' e.g. *stilla—stilel* 'stars' [14, 22].

Both systems have been and are still productive. The productivity can be witnessed by borrowings from Non-Semitic languages that pluralize in either the broken or sound pattern. For example, the Italian *delfino* has been borrowed as *denfil* in Maltese and pluralizes non-concatenatively as *dniefel* 'dolphins' [23]. Within the group of sound plural suffixes, some suffixes have been borrowed from Non-Semitic languages. This is the case for plural *-s*, such as in *kejks*

'cakes' from English, or the Sicilian-derived sound plural suffix -i, like in *fjuri* 'flowers'. Especially the suffix -i is reported as the most frequent sound plural suffix of the Maltese language [14, 17, 22]. Other suffixes are part of the Semitic heritage, such as -in and -iet, e.g. *barranin* 'aliens/foreigners' and *kelmiet* 'words', which are reported as having a straightforward Arabic origin [17].

While the broken plural system is likely to be reflected in phonotactics very specific to Semitic languages, the case is less clear in the sound system with its different suffixes whose word-final vowel and consonant sequences may potentially be present in both Semitic and Non-Semitic languages. Concerning these two different morphological classes, [24, 25] demonstrated that the two-layer network employed in the present study is very well capable to identify them on the basis of the phonotactic cues of the nouns. This potential impact on the classification brings us to the hypotheses of our present study.

## The present study

In the present study, we use modern Tunisian Arabic and modern Italian as representatives of Maghrebian Arabic and Sicilian Italian, the two languages which initially contributed to forming modern Maltese. Given that Tunisian Arabic and Italian also differ systematically in their phonotactics, and given that Maltese is uniformly regarded to be a descendant of both languages, we can state the following research hypothesis (RH):

**RH**: A neural network trained to classify Tunisian Arabic and Italian word forms as either Semitic or Italo-Romance based on the words' phonotactics should also be able to classify Maltese nouns that originate from Tunisian Arabic as Semitic and Maltese nouns that originate from Italian as Non-Semitic.

We will investigate to what degree the phonotactic classification of Maltese nouns as Semitic or Non-Semitic depends on their origin. We expect that the network will classify nouns of Semitic origin with a high accuracy. However, the question is whether this also holds true for nouns of a Non-Semitic origin. Italian (and English) words in Maltese are actually considered to be borrowings into an Arabic language [18] and it is established that borrowed words typically undergo adaptation to the target languages' phonology [26, 27]. From this follows that the phonotactic characteristics of Italian borrowings into Maltese may have become less similar to Italian and more similar to Arabic over time [28, p. 273]. We therefore expect that a noticeable percentage of nouns of Non-Semitic origin will also be classified as Semitic. Another aspect concerns the morphological class of Maltese singular and plural nouns: since broken plurals display specific phonotactic patterns, and this phonotactic pattern is most likely to be found in Semitic languages, we expect them to be classified as Semitic as well, independently of their origins. By contrast, the classification of sound plurals should depend on their origin, with Semitic nouns being classified as Semitic and Non-Semitic nouns classified as Non-Semitic.

Having described the background of our study as well as our hypothesis, we turn to our investigation. In the following section, we will first describe the data and networks used in our study before we report the results of our classification attempts. In our discussion, we turn our attention to the consequences of our research, specifically for the research on the Maltese typology.

## Methods

It is not surprising that the complex history of Maltese has sparked studies that aim to compare Maltese to Semitic and Non-Semitic languages. For example, [4] investigated the similarity of

Maltese to other Romance and Semitic languages by means of the *World Atlas of Language Structures* (WALS). Specifically, he used Spanish and Egyptian Arabic as representatives of the Romance and the Semitic language family. WALS lists a total of one hundred features distributed across categories such as phonology, morphology, nominal categories, nominal syntax, verbal categories, word order, simple clauses, complexity of sentences and the complexity of the lexicon. In six of these categories, Maltese was more similar to Egyptian Arabic than to Spanish. In two (morphology and lexicon) it was the same as Spanish and Egyptian Arabic. In one, word order, it was more similar to Spanish. [4] thus concluded that Maltese has to be classified as more similar to a Semitic language than to an Italo-Romance language. A point in Comrie's analysis that is of special interest for the present study is that of the Maltese lexicon: it is equally similar to Spanish and to Egyptian Arabic.

This kind of analysis undoubtedly depends on the material and features chosen for the comparison. For example, the computational studies discussed in the introduction typically use the cognates in the Swadesh list to classify a language as belonging to a specific language family. The Swadesh list that is part of the *Automated Similarity Judgment Program* (ASJP) [29] provides translations for 40 concepts in 5,590 Distinct ISO 639-3 languages. 38 of the 40 Maltese items in the Swadesh list, however, i.e. an overwhelming 95%, are of Semitic origin. Only 2 are Non-Semitic. On the basis of such a (small) data set, Maltese would would be classified to be most similar to Semitic languages. By contrast, if the list of Maltese nouns employed by [22, 24] was selected, the result would be completely different, since in their list 2621 out of 6511 nouns, i.e. only 26.7%, are of Semitic origin. From this perspective, Maltese would exhibit stronger characteristics of Non-Semitic languages. However, we decided not to classify the origin of Maltese words simply on the basis of their established origin, but on the basis of the structural similarity that Maltese words have to a Semitic and a Non-Semitic language. We gauged this similarity by training a two-layer network to classify Tunisian Arabic and Italian nouns as Semitic or Non-Semitic on the basis of their phonotactics, and having the trained network classify Maltese words as belonging to either of the two language families. With this background in mind, we describe our data and network approach in the following sections.

## The data

We used data from Tunisian Arabic and Italian corpora to train our network. For Tunisian Arabic, we took all word forms tagged as nouns from [30], a semi-automatically collected corpus of Tunisian Arabish data. Arabish is the romanization of Arabic Dialects for the purpose of messaging in social networks or chats. Thus, the corpus gave us the advantage of using a romanized Arabic script that contains vocalic information.

After the removal of unclear data, we obtained in total 2347 nouns. For Italian, we used the `subtlex-it` corpus downloaded from http://crr.ugent.be/programs-data/subtitle-frequencies.

We wanted to establish that our network is actually sensitive to phonotactic differences between two languages, and how the sensitivity varies if a different language would be used. Accordingly, we contrasted our Tunisian Arabic and Italian network with a network trained on Tunisian Arabic and German. In both cases, the network learns to discriminate between a Semitic and a Non-Semitic language. However, in the case when the network is trained with German, it is presented with a language that did not contribute to the Maltese lexicon. We therefore expect that a network will yield higher classification accuracies when it is trained on Tunisian Arabic and Italian in contrast to when it is trained on Tunisian Arabic and German. We extracted German words from the German `CELEX` corpus [31]. Both the Italian corpus and the German corpus are immensely larger than our Tunisian Arabic corpus. Using 2347

Tunisian Arabic nouns in training while using all words from `subtlex-it` and `CELEX` (whose numbers range in the tens of thousands) would heavily bias the neural network towards Non-Semitic. In order to obtain a relatively unbiased network, we provided an equal number of Semitic and Non-Semitic words to the network. Accordingly, we took a random sample of 2347 nouns from `subtlex-it` and from `CELEX`. To train the network, nouns from the Tunisian corpus were uniformly tagged as `semitic`, whereas Italian or German nouns were uniformly tagged as `non-semitic`.

We tested our trained networks with Maltese nouns. Our Maltese data set consists of 6511 nouns taken from [32]. The nouns in this data set are annotated for several properties such as grammatical number, CV structure or concreteness. For our purposes these properties were the origin of the nouns (Semitic vs. Non-Semitic) and their grammatical number including their morphological pattern (singular vs. broken plural vs. sound plural). Fig 1 displays the proportion of singular, broken and sound plural nouns within the two groups of origin in our Maltese data set:

Of the 6511 nouns in the data set, 3856 nouns have a Non-Semitic origin and 2621 have a Semitic origin. Within the group of Non-Semitic nouns (bars on the left), 49.6% are singulars (1913 words). 43.9% of the nouns in this group (1694 words) show a sound plural suffix and 6.5% of the nouns (249 words) have a broken plural pattern. Within the group of Semitic nouns (bars on the right), 46.4% of the nouns (1217 words) are singulars, 29.1% show a sound plural (763 words) and 24.5% have a broken plural form (641 words).

Of the remaining 34 of 6511 nouns that are not displayed in Fig 1, 15 words have a dual form, 8 words have a so-called double plural (a combination of a broken and a sound plural form), 6 have a suppletive plural, such as *mara—nisa* 'women', and 5 have an unclear origin. Due to the fact that these nouns cannot be unambiguously assigned to the classes `sound` vs. `broken` or `semitic` vs. `non-semitic`, they are not taken into consideration when we report the results of this study.

From Fig 1 it follows that a noun that is classified as having a Non-Semitic origin will be very likely pluralized with a sound plural suffix in Maltese. However, there is a small group of

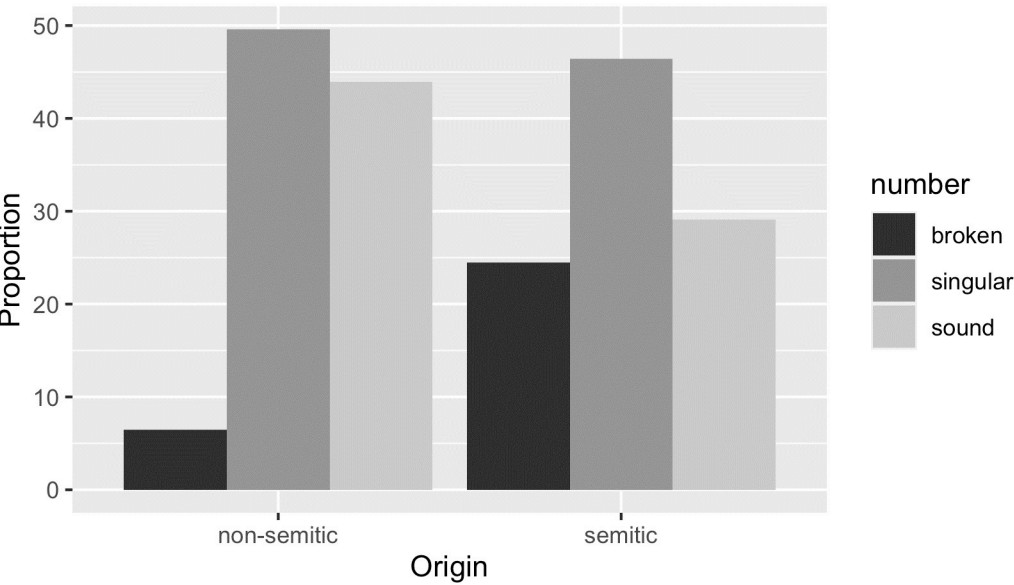

**Fig 1. Proportion of singular, broken and sound plural nouns in the Maltese data set classified as Semitic vs. Non-Semitic.**

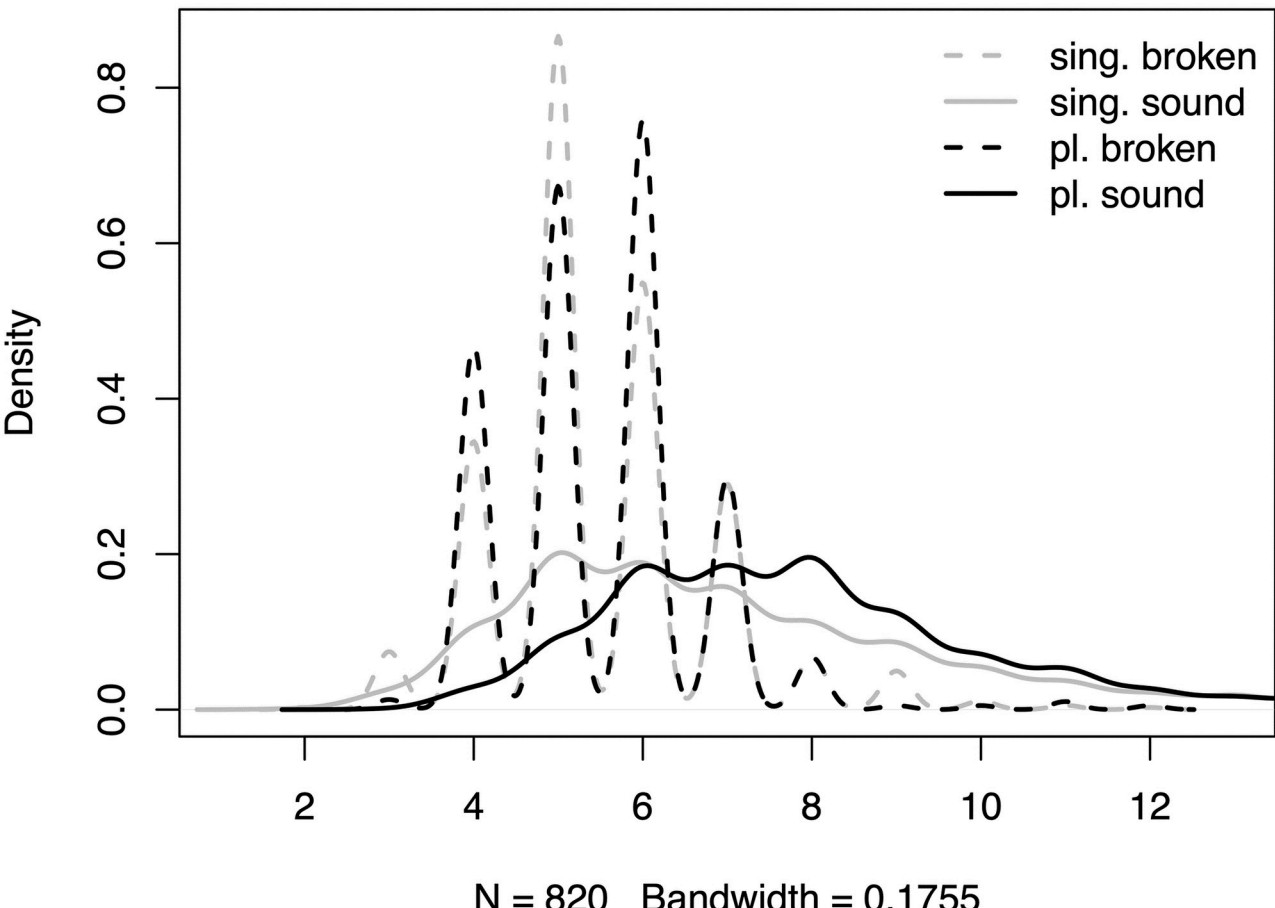

**Fig 2. Density plots of word length in the Maltese data.**

nouns in our data set that show a broken plural pattern instead (dark grey bar, group on the left). Thus, a noun with a Non-Semitic origin does not necessarily need to be pluralized in a concatenative way, instead there is a small number of words that can be pluralized with a broken plural pattern.

Turning to the nouns with a Semitic origin, we find the highest number of broken plural words (dark grey bar, group on the right) in this group. However, we also discover a considerable number of sound plurals within this group (light grey bar, group on the right). Considering that sound plurals make up the majority of plural forms in the Maltese language, there is no reason to worry about a bias during classification [14, 22, 24, 33].

Additionally, we checked our Maltese data set for word length since the length might have an influence on the classification of the network. For that, we decided to distinguish singulars into singulars that have a sound plural form and singulars that have a broken plural form (henceforth labelled as `singular sound` vs. `singular broken`). Fig 2 illustrates the distributions of word length in the Maltese data depending on morphological class.

We find that the length distributions for broken singulars and broken plurals strongly over-lap (mean length for both = 5.5). Words in the sound singular class are systematically longer (mean length = 7.1) and words in the sound plural class are longest (mean length = 8.3). The latter finding is not surprising in light of suffixation in the sound plural class.

Finally, in order to achieve a cross-language comparison, we created a quasi-phonetic tran-scription (based on SAMPA) for all word forms from all languages. Obviously, the process of creating a uniform quasi-phonetic transcription for languages from various language families with a different phoneme inventory made simplifications necessary, e.g. we ignored vowel duration in Tunisian Arabic.

Since a detailed list would use up a lot of space, we have provided the transcription in our supplementary materials: https://osf.io/r9k5n/, in addition to the R-code necessary to train our networks and perform our analyses.

## Input-Output structure

The network employed is a simple two-layer neural network that is trained to classify out-comes on the basis of provided cues using the Danks Equilibrium Equations [34] implemented in the Naive Discriminative Learner package for R [35, 36].

The present study represents a first attempt in using the Naive Discriminative Learner (NDL) for a cross-language modelling procedure WITHIN a language. The decision to use NDL as opposed to other possible language models and networks was based on several reasons. First, one of the aims of this study is to test the predictiveness of Tunisian and Italian phono-tactics for morphological functions in Maltese. For that, a network that uses a form-to-form mapping is necessary. NDL provides such a simple architecture in directly relating cues to out-comes [36]. Second, to be able to compare three very different languages on a morphological level, a network is necessary that is able to handle different morphologies (see [24] for a discus-sion why a model like the MGL by [37] does not work for non-concatenative morphology). Previous work by [24] has successfully shown that noun data from Maltese, combining conca-tenative and non-concatenative morphological patterns, can accurately be predicted with NDL despite the complexity of the noun system.

For the first part of our study, we trained two NDL networks. The first network was trained with data from Tunisian Arabic and Italian, the second network was trained with data from Tunisian Arabic and German. Based on previous research by [24] we used 2-phones as cues in all our networks. As outcomes, we used the language families semitic vs. non-semitic. Table 1 illustrates our cue-to-outcome structure.

The first column of Table 1 displays a singular noun. In the second column, the word forms are divided into 2-phone chunks that serve as cues in our networks. For illustratory purposes, the 2-phones are given in Latin script and do not represent our quasi-phonetic transcription. The third column contains a gloss for each word form. Finally, the rightmost column illus-trates the outcome for each word form: semitic vs. non-semitic.

**Table 1. Examples for cue-to-outcome structure for all languages.** Rows 1-3 represent the training data, row 4 illustrates the Maltese test data with Semitic being the correctly classified outcome.

|  | word form | 2-phone cues | gloss | outcome |
|---|---|---|---|---|
| Tunisian | basla | #b_ba_as_sl_la_a# | 'onion' | Semitic |
| Italian | serpe | #s_se_er_rp_pe_e# | 'snake' | Non-Semitic |
| German | Abend | #a_ab_be_en_nd_d# | 'evening' | Non-Semitic |
| Maltese | kelma | #k_ke_el_lm_ma_a# | 'word' | to be classified |

For a successful prediction, the network needs to classify the Maltese word *kelma* 'word' that is given in the last row of the table as Semitic based on the training on Tunisian and Italian or German. If the network instead classifies the word *kelma* as Non-Semitic, this would be counted as an unsuccessful prediction.

Our training data for Tunisian Arabic contained 549 unique 2-phone types; Italian had 341 unique 2-phone types, and German had 663 2-phone types. In these 2-phone types, 61% of the Tunisian Arabic 2-phones could be found in the Italian 2-phones; by contrast, 38% of the Italian 2-phones could be found in the Tunisian Arabic 2-phones. The test data set of Maltese contained 550 unique 2-phone types. The network trained on Tunisian Arabic and Italian was able to use 67% of the 2-phone types for the classification of Maltese words. The remaining 33% were treated as missing cues and disregarded during classification.

### Evaluating language similarities

In the introduction of this study, we highlighted the importance of Tunisian Arabic and Italian for the Maltese language system. To confirm empirically that Maltese is closer to Tunisian and Italian in terms of its phonotactic and phonological characteristics than to German and to justify the inclusion of the latter in the present study, we assessed the languages' similarity. We do this on the basis of the Swadesh list, in spite of its shortcomings, as it provides us with concept based translations. The Swadesh lists that are available for Maltese, Tunisian and Italian in the ASJP database contain 40 words for each language [38]. For each language, several options were available in the ASJP database. We chose the lists labeled `Italian`, `Tunisian Arabic Maghrib` and `Maltese` for this study. As mentioned above, for Maltese, only two of these words are Non-Semitic, namely *persuna* 'person' and *muntanja* 'mountain'.

We opted for the average normalized Levenshtein distance as a measure of similarity between the languages [29]. To obtain the average normalized Levenshtein distance, first, the Levenshtein distance between all words from $L_1$ and $L_2$ is calculated. For each $L_1$-$L_2$ word pair, the Levenshtein distance is then normalized by dividing each distance by the number of characters of the longer of the two words in the word pair that is to be compared. Finally, the average Levenshtein distance between $L_1$ and $L_2$ is obtained by taking the mean of the sum of all normalized Levenshtein distance values of the pairwise word comparison.

Fig 3 displays the average normalized Levenshtein distance between Maltese-Tunisian, Maltese-Italian and Maltese-German.

As expected, we find the greatest distance between Maltese and German with a distance of 0.94. For Maltese and Italian, the distance is at 0.87. Finally, we observe the distance of 0.57 between Maltese and Tunisian, validating the classification of Maltese as a Semitic language. Remember that the Swadesh list for Maltese contains a high proportion of word forms having a Semitic origin, 95% (38 of 40 word forms). Accordingly, we have tested the phonological similarity between the languages with the material at hand.

Based on these language similarity results, we conclude that including German as an unrelated language for the evaluation process is a reasonable approach. In the following, we turn our attention to the classification results of our networks.

## Classification of origin

### General classification

Table 2 reports the overall accuracies of the two networks trained on Tunisian Arabic + Italian and on Tunisian Arabic + German. Both networks were tested on the original data (accuracy training data) and on Maltese nouns (accuracy Maltese).

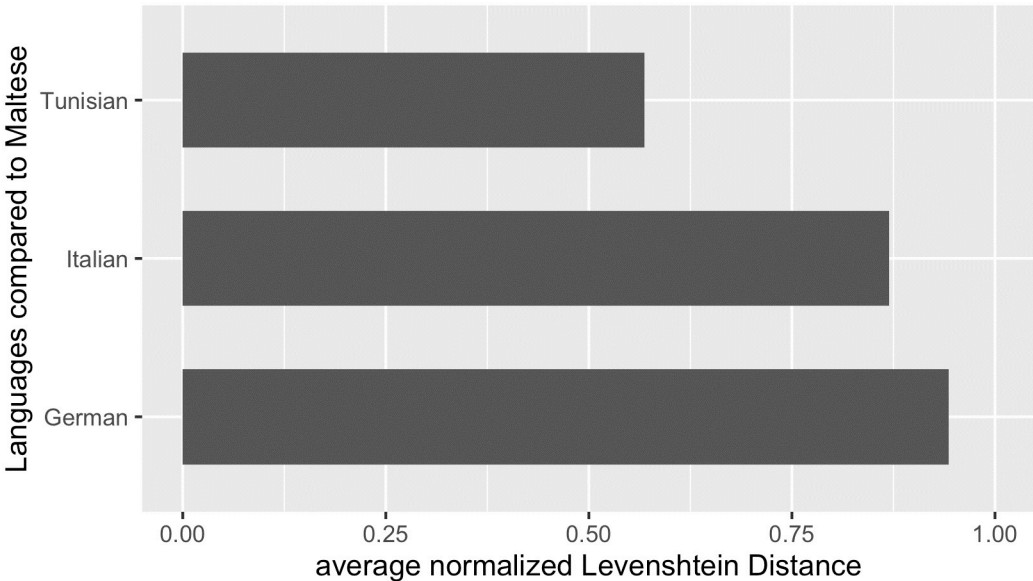

**Fig 3. Language similarity operationalized as average normalized Levenshtein distance between Maltese-Tunisian, Maltese-Italian and Maltese-German (based on the ASJP database by [38]).**

The network trained on Tunisian Arabic + Italian achieved an overall classification accuracy of 97.40%. This indicates that the network is very capable to use individual word forms to classify the two languages as Semitic vs. Non-Semitic. When the network was required to classify Maltese nouns as Semitic vs. Non-Semitic, the classification accuracy was at 70.76%. This percentage is higher than chance (50%), indicating that even Maltese words that are unknown to the network, can still be classified with a high accuracy.

The network trained on Tunisian Arabic + German achieved an overall classification of 97.15%, while the accuracy dropped to 61.09% when tested with Maltese. Since the classification accuracy is lower than for the network trained on Tunisian Arabic + Italian, the lower accuracy confirms our research hypothesis: the network is sensitive to the phonotactics of individual languages and can use phonotactic variation to classify them.

Recall that our intention is to assess a more fine-grained picture of family relations among languages. Accordingly, in the next section, we inspect how the classification of Maltese nouns as Semitic vs. Non-Semitic varies depending on their morphological function.

## Effect of morphological classes

Remember that we have argued that Non-Semitic nouns had to be adapted to the Semitic framework when they were borrowed into Maltese [28 p. 273]. Hence, we hypothesized that nouns of Non-Semitic origin should be classified as Semitic with a lower probability than

**Table 2. Global accuracies of the two NDL models trained on Tunisian and Italian nouns or trained on Tunisian and German nouns; tested on Maltese nouns.**

|  | Tunisian + Italian | Tunisian + German |
|---|---|---|
| accuracy training data | 97.40 | 97.15 |
| accuracy Maltese | 70.76 | 61.09 |

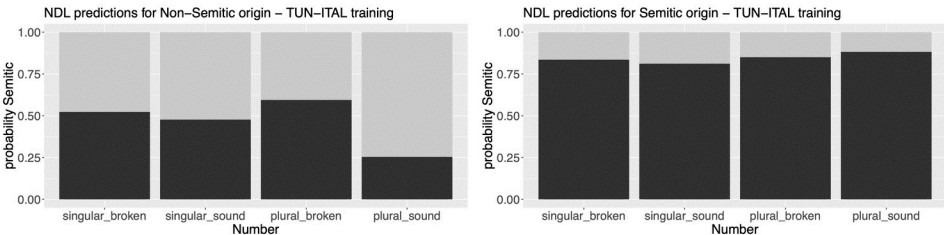

**Fig 4.** Classification probability (y-axis) for the Non-Semitic (left column) vs. Semitic group (right column) of Maltese nouns classified as Semitic by the Tunisian Arabic + Italian network. The x-axes show the morphological classes (singular broken vs. singular sound vs. plural broken vs. plural sound). The color coding indicates the predicted classifications of the model: dark grey = `semitic`, light grey = `non-semitic`.

those of Semitic origin (when tested with the Tunisian Arabic + Italian network). Fig 4 illustrates that this is exactly the case.

Ignoring the differences between morphological classes, we find that nouns with a Non-Semitic origin show an overall smaller classification probability as Semitic than those with a Semitic origin (39.0% vs. 84.6%, $\Delta$ = 45.6%, z-value = 34.34, p < 0.001). Whenever we present statistics, we used a generalized linear model, with family = binomial, to estimate the significance of the difference between two conditions. The dependent variable was a binomial variable, with TRUE = Semitic and FALSE = Non-Semitic.

Next, we will inspect the classification in light of its predictability for the different Maltese morphological classes. We have pointed out previously that some Non-Semitic nouns also show a broken, i.e. Semitic, plural pattern. The reason for this phenomenon may be that these singulars may actually have a phonotactic structure that is similar to Semitic phonotactics. Accordingly, we hypothesized that singulars with a broken plural will be classified as Semitic with a higher probability than singulars that have a sound plural. Critically, we expected a similar effect for plurals.

In Fig 4, we observe that singulars with a broken plural (singular_broken = 52.3%) have a slightly higher classification probability as Semitic than those with a sound plural (singular_sound = 47.7%). However, when tested with a generalized linear model, we found that the difference was not significant ($\Delta$ = 4.6%, z-value = -1.333, p = 0.183). This means our expectations were not corroborated in the singulars. By contrast, it was corroborated in the plurals. Plurals with a Non-Semitic origin have a higher classification probability as Semitic when they show a broken pattern than when they show a sound pattern (59.4% vs. 25.4%, $\Delta$ = 34%, z-value = 10.38, p < 0.001). This finding demonstrates that nouns of Non-Semitic origin have obtained phonotactic characteristics that are more similar to Semitic than to Non-Semitic languages.

Turning our attention to nouns with a Semitic origin, we find no significant difference between broken and sound singulars (83.5% vs. 81.1%, $\Delta$ = 1.4%, z-value = 1.127, p = 0.26), and no significant difference between broken and sound plurals (85.0% vs. 88.2%, $\Delta$ = 3.2%, z-value = 1.748, p = 0.08). In other words, across all morphological classes, nouns with a Semitic origin are uniformly classified as Semitic.

## Importance of consonant and vowel information

### Background

In the previous sections we have shown that our networks are capable to successfully classify the origin of Maltese nouns based on phonotactics. However, the question arises whether

Maltese nouns can also be classified on the basis of pieces of information other than the identity of vowels and consonants. This question is motivated by debates about Maltese grammar and what processes best explain the transformation of singulars into plurals in a grammatical system that consists of non-concatenative and concatenative forms.

In this regard, the most researched Semitic language is Arabic. [39], within their influential theory of Prosodic Morphology, argue that in Arabic, the CV template is a major factor in the productivity of noun plural patterns [40, 41].

The CV template is defined as one of three morphemes that a Semitic word is composed of [42], with the consonantal root and the vowel melody being the other two morphemes. A Semitic noun, e.g Arabic kita:b 'book' is described as consisting of the consonantal root $\sqrt{ktb}$, which expresses the meaning 'to write', and a melody of vowels (*i* and *ā*), that expresses the grammatical number singular. To build a plural form, both, consonants and vowels, are mapped onto the CV skeleton CVCVC to obtain *kutub* 'books'.

This root-and-pattern approach receives support from a computational study by [40] who report a better classification accuracy of Arabic broken plurals when the CV template is considered in the classification process of their model.

[22], on the other hand, investigated the transformation of singulars into plurals in Maltese. They conducted a production experiment in which Maltese participants had to pluralize existing and novel words. On the basis of the pluralization patterns produced by the participants, they concluded that pluralization in Maltese is best explained through a process of analogy, not the mapping onto a CV pattern: plurals for novel words are created on the basis of phonological similarity to existing forms and the frequency of pluralization patterns. [24, 25] have corroborated this finding by modeling the pluralization process with a simple two-layer network which predicted the plural class of singular nouns only on the basis of its phonological word form. The network thus learned to exploit systematic differences between the word forms in the broken and sound group. This finding indicates that the nouns in these two groups differ systematically in their phonotactics.

The question thus arises, to what degree abstract representations of vowels and consonants are useful to classify Maltese nouns as Semitic or Non-Semitic. Previous research on Semitic languages, such as the Prosodic Theory of Non-concatenative Morphology by [41], highlighted the importance of the CV skeleton for Semitic languages such as Arabic. Whether or not a similar assumption may apply to Maltese is under debate. For example, [43] show priming effects of the consonantal root in an auditory masked priming experiment, supporting the importance of a CV-skeleton for Maltese verbs. By contrast, [24] report no effect of the CV skeleton when they modeled the classification of Maltese noun plural classes. However, [24] had not investigated origin as a predictor for the classification of nouns to plural classes but the effects of individual sound plural suffixes and broken plural patterns. Thus, while the CV template might not have an influence on the classification of pluraliziation patterns, given that the CV skeleton is considered an inherent property of Semitic languages, it is possible that it has an influence on the classification of Maltese nouns as Semitic or Non-Semitic.

This brings us to the hypotheses for the second analysis. We will test the informativity of consonants and vowels, and the informativity of the CV skeleton for the two pluralization classes in Maltese by masking either consonants as C, vowels as V or both as CV. We expect masked input to be sufficiently informative about either language family and thus allow the network to classify the modulated input as either Semitic or Non-Semitic above chance ($> 50\%$). However, given that consonants are of utter importance in non-concatenative morphologies [44, 45], we expect a lower classification accuracy when consonants are masked in

contrast to when vowels are masked. When both types of phones are masked, we expect the lowest classification accuracy if the CV structure does not play a role for Maltese.

## Material

To test the importance of consonantal and vowel information for the classification of Maltese nouns, we trained three additional networks to classify Tunisian Arabic and Italian nouns as Semitic or Non-Semitic.

In the data provided to the first network, we masked consonantal information by uniformly coding all consonants in the cues as C, e.g. Italian *stoffa* 'fabric' → *CCoCCa*. In the second network, all vowels were masked as V, e.g. Italian *stoffa* 'fabric' → *stVffV*. In the third network, both consonants and vowels were masked, resulting in a CV skeleton for each word form in the training data, e.g. Italian *stoffa* 'fabric' → *CCVCCV*.

If the CV skeleton does indeed play a role for the classification of the origin of Maltese nouns, we expect a high classification accuracy for a network that was trained on both masked consonants and vowels. We expect this specifically for nouns of Semitic origin, while nouns of Non-Semitic should exhibit some uncertainty due to the aforementioned language changes.

In the case of masked consonants only, we expect a lower classification accuracy in comparison to masked vowels only. This is for three reasons. First, consonants have been demonstrated to play an important role during lexical processing of languages [46]. Second, as mentioned above, consonants are especially important in non-concatenative morphologies due to carrying specific meanings [44, 45]. Third, the phoneme repertoire of the languages under investigation contain more consonants than vowels, making the former more informative for distinguishing between word forms.

## Results

Table 3 illustrates the classification results for the masked nouns. When the networks were required to classify the training data, they showed excellent classification accuracies when only vowels or consonants are masked. The classification accuracy was slightly lower when both phone classes are masked. This indicates that the sequence of consonants and vowels is sufficiently informative about each language family. The classification accuracy, however, is strongly reduced when the networks were required to classify Maltese data. Yet, the classification is still above chance ($> 50\%$), indicating that the CV-skeleton presents sufficient information about language families. Crucially, when only consonants were masked, classification accuracy was lower than when only vowels were masked. As hypothesized, this indicates that consonants play an important role in the classification of Maltese.

Next, we turn our attention to the details of classification depending on the individual morphological classes in Maltese. Fig 5 illustrates the equivalent findings for the nouns with masked phones. In all three networks, the probability of Maltese nouns that were classified as Semitic was higher for words of Semitic origin (right column) than for words of Non-Semitic origin (left column; for all three differences between Semitic and Non-Semitic: z > 15.00, p < 0.001).

**Table 3. Global accuracies of the three NDL models trained on Tunisian and Italian nouns with masked C, masked V or masked CV as cues; tested on Maltese nouns.**

|  | masked C | masked V | masked C+V |
|---|---|---|---|
| accuracy training data | 94.14 | 94.12 | 78.29 |
| accuracy Maltese | 58.20 | 65.18 | 60.50 |

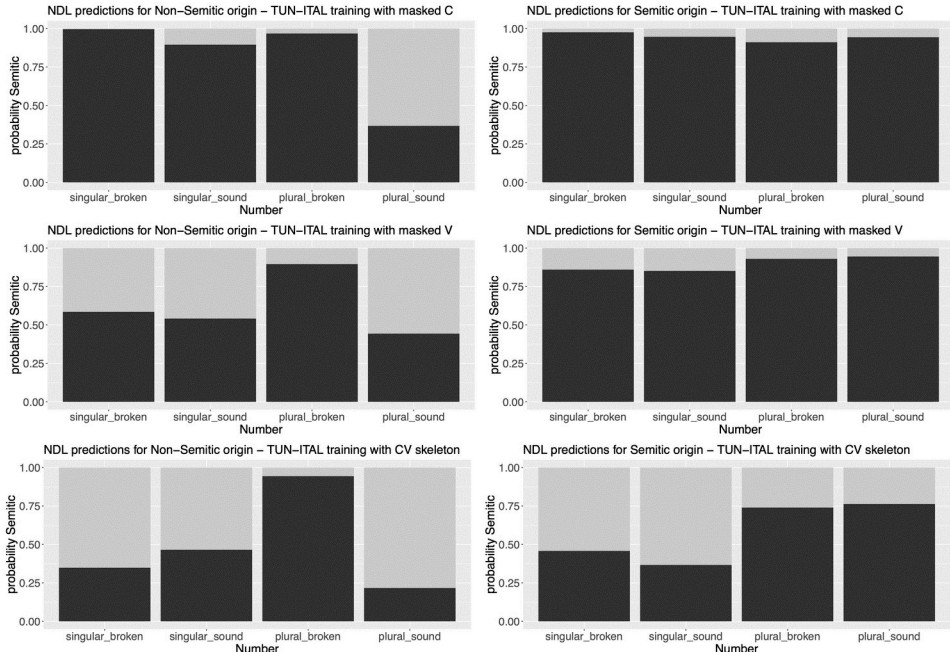

**Fig 5.** Classification probabilities (y-axes) for the Non-Semitic (left column) vs. Semitic group (right column) of Maltese nouns classified as Semitic by the models trained on Tunisian-Italian with masked consonants, masked vowels and CV skeleton (rows).

Concerning the details of how the different masks affected classification, we first inspect the nouns of Non-Semitic origin (left column of Fig 5). When all consonants were masked as C (top row of Fig 5), we observe that singulars with both broken and sound plurals were systematically classified as Semitic (99.6% and 89.5%, respectively). This is also the case for broken plurals. Only sound plurals were reliably classified as Non-Semitic (difference between broken plurals and sound plurals: 96.8% vs. 36.7%, $\Delta$ = 60.1%, z-value = 10.884, p < 0.001). One potential reason for this finding is that sound plurals are marked by vocalic suffixes [22], such as *-i* or *-ijiet*, providing sufficient information about the word and accordingly about its membership to a language family.

We find the same significant difference between broken plurals and sound plurals (89.6% vs. 44.3%, $\Delta$ = 45.3%, z-value = 11.16, p < 0.001) when all vowels were masked as V and the network was provided with the identity of the consonants (left column in the middle row of Fig 5). By contrast, the classification of singulars is almost at chance (58.5% vs. 54.1%). This indicates that that the network is uncertain about the language family when the vowels are masked.

Finally, when consonants are masked as C and vowels as V, giving the network only the CV skeleton as input but hiding the phonemic identity (left column in the bottom row of Fig 5), all morphological classes but the broken plurals are classified as Non-Semitic above chance (difference between singulars with a broken and sound plural: 34.9% vs. 46.5%; difference between broken and sound plurals: 94.4% vs. 21.7%, $\Delta$ = 72.7%, z-value = 14.59, p < 0.001). This is an interesting finding that indicates that broken pluralization of nouns with a Non-Semitic origin actually follows Semitic phonotactics independently of the phonemic identity of consonants and vowels.

Next we turn our attention to the words with Semitic origin (right column of Fig 5). We observe a very high classification probability as Semitic when only the consonants or only the vowels are masked (top and mid row in the right column of Fig 5; we avoid reporting the statistics here for obvious reasons). Accordingly, the identity of vowels (in the case of masked consonants) or consonants (in the case of masked vowels) is still informative about the nouns' Semitic origin.

By contrast, when only the CV skeleton is provided to the network (bottom row in the right column of Fig 5), it is faced with greater uncertainty about the family class. Singulars are classified with an above chance as Non-Semitic while plurals are classified above chance as Semitic (40.7% vs. 75.2%, $\Delta$ = 34.5%, z-value = 17.777, p < 0.001). This finding indicates that the CV pattern is non-informative for singulars. By contrast, the CV skeleton in plurals of Semitic origin is very informative about the language family, as argued by the Prosodic Theory of Non-concatenative Morphology [1].

## Discussion

Due to its history of extensive language contact, Maltese grammar and phonology show similarities to Semitic and Italo-Romance languages. This similarity has been well documented in comparative and typological studies [4, 17, 47]. The present study extended these studies in two ways.

Our first extension was that we, in contrast to previous typological studies on Maltese, employed a computational approach. We made use of a simple two-layer neural network (NDL) [35, 36] to classify individual words (specifically nouns) as belonging to the either of the two language families. For this, we first trained the neural network to be able to classify Tunisian Arabic as belonging to the Semitic language family and Italian as belonging to the Italo-Romance, i.e. Non-Semitic language family, on the basis of the phonotactics of the presented words. Crucially, the network never had access to Maltese data in the training.

Subsequently, we had the trained network classify Maltese nouns as Semitic or Non-Semitic. Even though the network did not know many Maltese phonetic cues since they were not present in the original Italian and Tunisian Arabic training data, we found that the network was able to arrive at a classification with a relatively high accuracy. When the network was trained with Tunisian Arabic and German, i.e. a language that is considered to be unrelated to Maltese, the classification accuracies were lower. These differences in classification accuracy indicate that our network is capable to appropriately identify the different phonotactic structures of different languages and therefore language families, and use them to classify unknown words as belonging to either one of them.

Given that Maltese has an established group of Semitic and Non-Semitic words, we further inspected the classification accuracies depending on these groups. We found higher classification accuracies for the group of Maltese nouns that have a Semitic origin when compared to the group of nouns that have a Non-Semitic origin. Since the classification is based on the phonotactic characteristics of the words, this finding indicates that Semitic nouns in Maltese are more similar to their Semitic relatives than Non-Semitic nouns to their Non-Semitic relatives in terms of their phonotactic and phonological characteristics. This finding may be a direct result of language change. It is known that the characteristics of Non-Semitic words were adapted to the Semitic frame of Maltese when they were borrowed [28, p. 273]. These changes made the borrowed nouns less similar to their Non-Semitic origins, which is reflected in their higher classification as Semitic than as Non-Semitic.

Our second extension concerns Maltese grammar. When inspecting the classification of Maltese nouns, we took into account the nouns' morphological classes, specifically whether

they pluralize in a broken or a sound pattern. Clearly, there is a large overlap between the pluralization patterns and the nouns' origin, such that most Semitic nouns have a broken plural and most Non-Semitic nouns have a sound plural (see Fig 1). However, some Non-Semitic nouns have a broken plural and some Semitic nouns a sound plural. We investigated how this separation into different pluralization patterns affected the classification of Maltese nouns as Semitic vs. Non-Semitic. This separation did not have an effect on the classification of Semitic nouns, but a strong effect on the classification of Non-Semitic nouns. Specifically, Non-Semitic nouns with a broken plural were classified as Semitic with a higher probability than those with a sound plural. This indicates that the morphological class mirrors the phonotactic characteristics of the different language families.

In a next step, we tested the importance of consonantal and vowel information for the classification of Maltese nouns as Semitic or Non-Semitic. To do so, we ran three additional models with masked consonant information, masked vowel information and both masked consonants and vowels (the latter network represented a training on the CV skeleton).

In line with work by [46], the network showed the lowest classfication accuracy when consonantal information was masked during training, confirming that consonants play a more important role for lexical processing than vowels. When both consonants and vowels were masked, giving the network only the CV skeleton as input, we found different results for the group of Non-Semitic vs. the group of Semitic nouns. Within the Non-Semitic group all morphological classes, except for the broken plurals, were successfully classified as Non-Semitic, indicating that broken pluralization follows a more Semitic phonotactic structure. Within the Semitic group only the plurals broken and sound, but not the singulars, were successfully classified as Semitic, indicating that the CV skeleton is only informative for the origin of the Maltese plural classes.

Our results then do not only provide empirical evidence for the status of Maltese as being a language with an intertwined morphological system as a result of extensive language contact, they also serve as a base for future cross-language modelling studies. Clearly, Maltese represents a test case in which the typological origin of word forms is very well defined. However, the present approach can be used to identify the origin of borrowings, for example in German, or the identification of the origin for individual words with non-established origins in various languages.

## Acknowledgments

We want to thank our colleagues from the Heinrich-Heine-Universität Düsseldorf for their helpful comments on this article during a colloquium talk. A special thank you goes to Johannes Dellert for his valuable comments on previous versions of this paper and Michael Vella Bardon for proof-reading the article for us.

## Author Contributions

**Conceptualization:** Jessica Nieder.

**Data curation:** Jessica Nieder, Fabian Tomaschek.

**Formal analysis:** Jessica Nieder, Fabian Tomaschek.

**Investigation:** Jessica Nieder, Fabian Tomaschek.

**Methodology:** Jessica Nieder, Fabian Tomaschek.

**Project administration:** Jessica Nieder.

**Supervision:** Jessica Nieder.

**Validation:** Jessica Nieder, Fabian Tomaschek.

**Visualization:** Jessica Nieder, Fabian Tomaschek.

**Writing – original draft:** Jessica Nieder, Fabian Tomaschek.

**Writing – review & editing:** Jessica Nieder.

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
