## [Decision Letter · Decision Letter 0]

10 Feb 2023

PONE-D-23-02200Maltese as a Merger of Two Worlds: 

A cross-language approach to phonotactic 

classificationPLOS ONE

Dear Dr. Nieder,

Thank you for submitting your manuscript to PLOS ONE. After careful consideration, we feel that it has merit but does not fully meet PLOS ONE’s publication criteria as it currently stands. Therefore, we invite you to submit a revised version of the manuscript that addresses the points raised during the review process.

We look forward to receiving your revised manuscript.

Kind regards,

Simon Grima, PhD

Academic Editor

PLOS ONE

Additional Editor Comments:

Maltese as a Merger of Two Worlds: A cross-language approach to phonotactic classification

Abstract

1. The author has picked a unique topic, "PHONOTACTICS”, which is an area of phonology concerned with the analysis and description of the permitted sound sequences of a language and is the main focus of this paper. According to the abstract, the novelty lies only in using computational methods compared to hands-on comparative methods. Please clarify this novelty part.

2. “Moreover, 16 we find that the classification depends on whether a noun forms a sound or broken plural.” Please correct this line.

3. What are the findings not mentioned in the abstract?

4. In my opinion, Phonotactics refers to the principles according to which languages allow sound combinations and segment sequencing; I did not see any challenges related to Phonotactics have to do with the universal versus language-specific nature of phonotactic rules and preferences. Were there any? I didn't find any such review even on the same

5. Please focus on the research gap in the introduction. Strengthen the argument of your study by referring to a larger body of scientific literature, and clearly outline what is missing in the literature and what gap you wish to fill in.

Reviewers' comments:

Reviewer's Responses to Questions

**Comments to the Author**

1. Is the manuscript technically sound, and do the data support the conclusions?

Reviewer #1: Yes

Reviewer #2: Yes

2. Has the statistical analysis been performed appropriately and rigorously? 

Reviewer #1: Yes

Reviewer #2: Yes

3. Have the authors made all data underlying the findings in their manuscript fully available?

Reviewer #1: Yes

Reviewer #2: Yes

4. Is the manuscript presented in an intelligible fashion and written in standard English?

Reviewer #1: Yes

Reviewer #2: Yes

5. Review Comments to the Author

Reviewer #1: Thank you for opportunity this paper, very interesting for me. The paper provides empirical evidence for the status of Maltese as being a language with an intertwined morphological system as a result of extensive language contact, also serve as a base for future cross-language modelling studies. Clearly, Maltese represents a test case in which the typological origin of word forms is very well defined.

Reviewer #2: Maltese as a Merger of Two Worlds: A cross-language approach to phonotactic classification

Abstract

1. The author has picked a unique topic, "PHONOTACTICS”, which is an area of phonology concerned with the analysis and description of the permitted sound sequences of a language and is the main focus of this paper. According to the abstract, the novelty lies only in using computational methods compared to hands-on comparative methods. Please clarify this novelty part.

2. “Moreover, 16 we find that the classification depends on whether a noun forms a sound or broken plural.” Please correct this line.

3. What are the findings not mentioned in the abstract?

4. In my opinion, Phonotactics refers to the principles according to which languages allow sound combinations and segment sequencing; I did not see any challenges related to Phonotactics have to do with the universal versus language-specific nature of phonotactic rules and preferences. Were there any? I didn't find any such review even on the same

5. Please focus on the research gap in the introduction. Strengthen the argument of your study by referring to a larger body of scientific literature, and clearly outline what is missing in the literature and what gap you wish to fill in.

6. PLOS authors have the option to publish the peer review history of their article (what does this mean?). If published, this will include your full peer review and any attached files.

Reviewer #1: No

Reviewer #2: **Yes: **KIRAN SOOD

While revising your submission, please upload your figure files to the Preflight Analysis and Conversion Engine (PACE) digital diagnostic tool, https://pacev2.apexcovantage.com/. PACE helps ensure that figures meet PLOS requirements. To use PACE, you must first register as a user. Registration is free. Then, login and navigate to the UPLOAD tab, where you will find detailed instructions on how to use the tool. If you encounter any issues or have any questions when using PACE, please email PLOS at figures@plos.org. Please note that Supporting Information files do not need this step.<quillbot-extension-portal></quillbot-extension-portal>

---

## [Author Response · Author response to Decision Letter 0]

3 Mar 2023

1. The author has picked a unique topic, “PHONOTACTICS”, which is an area of phonology

concerned with the analysis and description of the permitted sound sequences of a language

and is the main focus of this paper. According to the abstract, the novelty lies only in using

computational methods compared to hands-on comparative methods. Please clarify this novelty

part.

Response: We have made this aspect of our paper stronger in the abstract and in the

introduction of this paper.

This is the new abstract:

“Maltese is a prime example of a language that emerged through extensive language contact,

joining the two linguistic worlds of Semitic and Italo-Romance languages. Previous

studies have shown this shared origin on the basis of hands-on comparative methods.

However, such approaches may be biased by the researchers perspective and the selected

material. To avoid this bias, we employed a naive computational method that classifies

words on the basis of their phonotactics. Specifically, we trained a simple two-layer neural

network on Tunisian and Italian nouns, i.e. the languages that Maltese emerged from.

We used the trained network to classify Maltese nouns based on their phonotactic characteristics

as either of Tunisian or of Italian origin. Overall, the network is capable of

correctly classifying Maltese nouns as belonging to either of the original languages. Moreover,

we find that the classification depends on whether a noun has a sound or broken

plural. Moreover, by manipulating the segment identity in the training input, we found

that consonants are more important for the classification of Maltese nouns than vowels.

While our results replicate previous comparative studies, they also demonstrate that a

more fine grained classification of a language’s origin can be based on individual words

and morphological classes.”

This is on page 2 of the new manuscript:

“In doing so, we employ a less biased method for the comparison of Maltese with 25

Semitic and Italo-Romance languages and simultaneously increase the amount of data

26 investigated. Accordingly, our study introduces a computational method based on 27

phonotactic classification to the study of the origin of Maltese. ”

2. “Moreover, 16 we find that the classification depends on whether a noun forms a sound or

broken plural.” Please correct this line.

Response: We replaced “whether a noun forms ...” with “whether a noun has...”.

3. What are the findings not mentioned in the abstract?

Response: We think that the reviewer is referring to our analysis concerning the informativity

of vowels and consonants at this point. We have therefore added a sentence

mentioning these results.This is the new abstract:

“Maltese is a prime example of a language that emerged through extensive language contact,

joining the two linguistic worlds of Semitic and Italo-Romance languages. Previous

studies have shown this shared origin on the basis of hands-on comparative methods.

However, such approaches may be biased by the researchers perspective and the selected

material. To avoid this bias, we employed a naive computational method that classifies

words on the basis of their phonotactics. Specifically, we trained a simple two-layer neural

network on Tunisian and Italian nouns, i.e. the languages that Maltese emerged from.

We used the trained network to classify Maltese nouns based on their phonotactic characteristics

as either of Tunisian or of Italian origin. Overall, the network is capable of

correctly classifying Maltese nouns as belonging to either of the original languages. Moreover,

we find that the classification depends on whether a noun has a sound or broken

plural. Moreover, by manipulating the segment identity in the training input, we found

that consonants are more important for the classification of Maltese nouns than vowels.

While our results replicate previous comparative studies, they also demonstrate that a

more fine grained classification of a language’s origin can be based on individual words

and morphological classes.”

4. In my opinion, Phonotactics refers to the principles according to which languages allow sound

combinations and segment sequencing; I did not see any challenges related to Phonotactics have

to do with the universal versus language-specific nature of phonotactic rules and preferences.

Were there any? I didn’t find any such review even on the same

Response: We absolutely agree with the reviewer’s statement here that phonotactics

refers to language specific combinations and sequences of sounds. However, we never intended

to investigate to what degree phonotactics may be universal or language specific.

Quite the contrary. In our study, we exploited the language specific distribution of sounds

and sound sequences to have Maltese nouns classified as either Semitic or Non-Semitic.

As we have repeatedly stated in our manuscript, our model learned the phonotactics of

Tunisian and Italian during the training and was able to predict the origin of Maltese

nouns based on their phonotactic characteristics. To make the role of phonotactics more

clear, we slightly changed the text of our research hypothesis and added “phonotactic classification”

in the following paragraph on page 4 of the new manuscript:

“RH: A neural network trained to classify Tunisian Arabic and Italian word forms as

128 either Semitic or Italo-Romance based on the words’ phonotactics should also be 129

able to classify Maltese nouns that originate from Tunisian Arabic as Semitic and 130

Maltese nouns that originate from Italian as Non-Semitic.”

“We will investigate to what degree the phonotactic classification of Maltese nouns as

132 Semitic or Non-Semitic depends on their origin.”

5. Please focus on the research gap in the introduction. Strengthen the argument of your

study by referring to a larger body of scientific literature, and clearly outline what is missing

in the literature and what gap you wish to fill in.

Response: Maltese is still a rather underresearched language, which is why the number

of studies about this topic is relatively small. This is why, in our humble opinion, we

think we have referenced all the relevant studies on Maltese comparatistic linguistics in

our manuscript, the most important one being Comrie 2009. Concerning the studies on

employing computational methods to compare languages, again, to our knowledge, since

this is a relatively new scientific field, there are only a few studies that have employed

this technique. We have mentioned those studies that have motivated us to perform the

present study on Maltese such as Jäger 2015, Jäger & Wichmann 2016, Jäger et al. 2017,

Jäger & List 2018 and List 2022. To our knowledge, the model that we are applying has

not been used before for a cross-language modelling procedure within a language. That is

why our study could also be described as being a contribution to this new scientific field.

Regarding the usage of the model for Maltese, we cited the two relevant papers employing

this technique for Maltese noun inflections (Nieder et al. 2021, 2022). However, if the

reviewer is able to point us to any studies they think are also highly relevant for the present

paper, we are of course happy to include them in our manuscript. In addition, we have

included the following sentence to our introduction to make the missing gap more apparent:

”In doing so, we employ a less biased method for the comparison of Maltese with Semitic

and Italo-Romance languages and simultaneously increase the amount of data employed.

Accordingly, our study introduces a computational method based on phonotactics to the

study of the origin of Maltese.”

---

## [Editor Report · Decision Letter 1]

3 Apr 2023

Maltese as a merger of two worlds:

A cross-language approach to phonotactic

classification

PONE-D-23-02200R1

Dear Dr. Nieder,

We’re pleased to inform you that your manuscript has been judged scientifically suitable for publication and will be formally accepted for publication once it meets all outstanding technical requirements.

Kind regards,

Simon Grima, PhD

Academic Editor

PLOS ONE

Additional Editor Comments (optional):

Reviewers' comments:

<quillbot-extension-portal></quillbot-extension-portal>

---

## [Editor Report · Acceptance letter]

6 Apr 2023

PONE-D-23-02200R1 

Maltese as a merger of two worlds: A cross-language approach to phonotactic classification 

Dear Dr. Nieder:

I'm pleased to inform you that your manuscript has been deemed suitable for publication in PLOS ONE. Congratulations! Your manuscript is now with our production department. 

Kind regards, 

on behalf of

Professor Simon Grima 

Academic Editor

PLOS ONE